# Delineation of Nitrate Reduction Hotspots in Artificially Drained Areas through Assessment of Small-Scale Spatial Variability of Electrical Conductivity Data

**DOI:** 10.3390/s22041508

**Published:** 2022-02-15

**Authors:** Maria Isabel Senal, Anders Bjørn Møller, Triven Koganti, Bo V. Iversen

**Affiliations:** 1Department of Agroecology, Aarhus University, Blichers Allé 20, 8830 Tjele, Denmark; anbm@agro.au.dk (A.B.M.); triven.koganti@agro.au.dk (T.K.); 2Aarhus University Centre for Water Technology (WATEC), Department of Agroecology, Blichers Allé 20, 8830 Tjele, Denmark

**Keywords:** nitrate reduction hotspots, electrical conductivity, unsupervised ISODATA clustering, Getis–Ord Gi* statistics, optimized hot spot analysis

## Abstract

Identification of nitrate reduction hotspots (NRH) can be instrumental in implementing targeted strategies for reducing nitrate loading from agriculture. In this study, we aimed to delineate possible NRH areas from soil depths of 80 to 180 cm in an artificially drained catchment by utilizing electrical conductivity (EC) values derived by the inversion of apparent electrical conductivity data measured by an electromagnetic induction instrument. The NRH areas were derived from the subzones generated from clustering the EC values via two methods, unsupervised ISODATA clustering and the Optimized Hot Spot Analysis, that highly complement each other. The clustering of EC values generated three classes, wherein the classes with high EC values correspond to NRH areas as indicated by their low redox potential values and nitrate (NO_3_^−^) concentrations. Nitrate concentrations in the NRH were equal to 13 to 17% of the concentrations in non-NRH areas and occupied 26% of the total area of the drainage catchments in the study. It is likely that, with the identification of NRH areas, the degree of nitrogen reduction in the vadose zone may be higher than initially estimated at the subcatchment scale.

## 1. Introduction

Approximately 90% of the nitrogen load in Danish coastal waters originates from diffuse sources, primarily through agriculture [1]. This situation challenges the achievement of nitrogen reduction goals under the European Union Water Directive, as the large heterogeneity in hydro-geochemical and biotic conditions causes large variations in the nitrogen export between different fields [2]. In addition, the scale at which the heterogeneity is assessed also influences estimates of the nitrogen export [3]. As a nutrient management strategy, a targeted, farm-specific approach that accounts for the heterogeneity of soil conditions is being devised [4].

As a basis for such an approach, the current national nitrogen model for Denmark [5] combines the statistical nitrogen leaching model with groundwater-surface hydrological and retention models. The model estimates the percentage of the total nitrogen that remains reduced as the water leaves the root zone and reaches the surface water systems. However, the model fails to account for locations at which reduction occurs, i.e., whether the reduction occurs within the same catchment or in another catchment area where the total nitrogen was measured [6]. Determination of the location and extent of nitrate reduction is important in formulating optimal and cost-effective strategies for nitrogen management [7].

Several studies have assessed the location and spatial heterogeneity of nitrate reduction through the identification of the redox interface location [8], characterization of the redox architecture [9], and multiple geological realizations [10], though this is mainly useful in the saturated zone. As for the extent of nitrate reduction in the unsaturated zone in the Danish context, studies are more limited [11]. Nitrate reduction in the unsaturated zone across different types of Danish soils has not been fully assessed as most studies are carried out under controlled laboratory conditions that are meant to describe the possible spatial and temporal heterogeneity under field conditions [11]. Denitrification potential [12,13,14], oxygen, organic carbon, and nitrogen species levels [13,15] are among the factors that have been used as variables for describing the heterogeneity of nitrogen reduction. Since these variables usually involve extensive sample collection and laboratory work, and are generally destructive, the sample sizes are often small and locations must be carefully selected to properly represent spatial characteristics [15,16]. Descriptions of physico-chemical properties tend to be inferred for large areas when sampling points are sparse and highly spaced, which could compromise the quality of soil heterogeneity assessment [17].

Electromagnetic induction (EMI) is a frequently employed proximal soil sensing technique to supplement time-consuming and labor-intensive conventional soil mapping for high-resolution spatial characterization of soil properties through measurement of the soil electrical conductivity (EC) [17,18,19]. The soil EC mainly relates to its physio-chemical properties. The salinity is the primary influential factor for soils with EC > 100 mS m^−1^, while other edaphic factors, such as texture, moisture content, bulk density, cation exchange capacity, and organic matter, contribute to the EC in non-saline agricultural soils (i.e., EC < 100 mS m^−1^; [18,20]). The dissolved solids within the water of the larger pores, the liquid phase, is the primary pathway for apparent soil conductance [21]. Nonetheless, contributions by the liquid–solid interface through the exchangeable cations associated with clay particles and the solid phase through direct contact of soil particles, may also be significant [21,22]. Hence, EMI has been regularly used for mapping the above stated soil properties, a few examples of which include salinity [23,24,25], texture [26,27,28], organic matter [29], cation exchange capacity [30], water content [31], and also, more importantly, for the delineation of management zones [32,33,34].

To briefly describe the working principle, the EMI sensors measure the apparent electrical conductivity (EC_a_) derived from the quadrature-phase response of the primary and secondary magnetic fields recorded by the receiver coils [35]. The secondary field is generated due to the eddy currents in the conductive material present in the subsurface that are induced by a transient primary field generated through powering the transmitter coil using an alternating current [36].

Recent technological advancements have led to the development of on-the-go measuring systems, which provide EC_a_ measurements up to multiple depths, making it feasible for a rapid collection of densely sampled data across large areas [17]. Subsequently, sophisticated inversion routines can be employed on these datasets to determine depth-specific estimates of EC, enabling the mapping of soil properties in three dimensions [23,24,30,37]. Mapping soil properties with EMI also gives an additional advantage of revealing complexities in the soil-landscapes patterns, particularly in disturbed, land-levelled landscapes (e.g., artificially drained agricultural areas), which could have been disregarded with methods relying on point measurements such as monitoring wells, core samples, and soil moisture probes [16,17]. Furthermore, the direct current resistivity, an electrical method counterpart, either employed as an on-the-go measuring system or as electrical resistivity tomography, i.e., by injection of a series of electrodes into the ground, can provide similar EC_a_ measurements and depth-specific EC estimates [30,38,39]. However, as all the electrical (direct current resistivity, induced polarization, and self-potential) methods necessitate direct electrode contact with the ground, they are more suitable for the plot scale investigations, and their applicability is severely compromised when the spatial extent of the area being surveyed is upscaled to the catchment level.

In this relation, to discern the soil nitrogen status at catchment level, Senal et al. [40] employed EMI sensor measurements and showed that variation in soil EC reflected patterns in potential nitrogen removal areas, as areas with higher EC often displayed reducing conditions. The authors linked this finding to clay contents, which reduce the hydraulic conductivity of the subsoil and lead to waterlogged conditions, as high clay contents and moisture increase the soil EC [17,27]. Clay contents have also been associated with the presence of anoxic microsites, which are potential sites for active nitrate reduction [41,42].

However, in order to make this finding operational for management purposes, it is necessary to delineate areas with high EC in geographic space and to determine the nitrogen removal capabilities of these areas. For this purpose, clustering algorithms can be a solution with a considerable potential.

The unsupervised ISODATA (Iterative Self-Organizing Data Analysis) clustering algorithm [43] is a widely used clustering method which measures the means within a data space and iteratively generates clusters from the remaining pixels using minimum distance techniques. The process iterates until a maximum number of iterations has been reached. It is a relatively straightforward and computationally inexpensive clustering technique for delineating zones for management purposes [44,45]. However, it often depicts abrupt changes in soil properties at boundaries between subzones, which is unlikely in actual soil conditions [45], and relies on maps of spatially interpolated variables generated either through kriging or co-kriging techniques.

A promising alternative clustering technique is based on Getis–Ord Gi* statistics [46] which evaluates the statistical significance of the value at a point observation based on the context of the neighboring point observations. A high value is only considered significant when the neighboring features also have significantly high values. A cluster of these significantly high values may then be regarded as a hot spot. The Getis–Ord Gi* statistics has been widely used in studies pertaining to transportation [47,48], public health [49,50,51], species distributions [52,53,54], etc., but its application for soil studies is still limited, most likely due to the limited availability of spatially dense data such as EC.

The hypothesis is that subzones with high EC values are potential nitrate reduction hotspot (NRH) areas, which can be determined through their low average nitrate (NO_3_^−^) concentrations and redox potential values (Eh). The use of Optimized Hot Spot Analysis for generating the subzones is based on the assumption that the EC values have an acceptable density for this clustering method. By identifying the potential NRH in the soil layers above the drainage pipes, we also characterize the spatial heterogeneity of nitrate reduction in the unsaturated zone, an area that is still lacking understanding in its heterogeneity and processes. This will support the efforts for creating targeted, farm-specific nutrient management strategies. It also is worth a mention that, though induced polarization can be useful to understand the physio-chemical processes and the relationship between the nitrate reduction hotspots and the soil EC [55,56], an EMI sensor was chosen in this study, as it is optimal for surveying large catchment scale areas and acquiring densely sampled data.

This study aims to delineate potential nitrate reduction hotspot (NRH) areas within the vadose zone above the drainage pipes. To achieve this, we have two objectives: (1) to utilize EC values acquired through the EMI technique to generate subzones with signature nitrogen and redox profiles, and (2) to generate these subzones through clustering of EC values via two different methods: unsupervised ISODATA clustering and Getis–Ord Gi* statistics (utilized through the Optimized Hot Spot Analysis). Comparison of clusters generated via the two methods were conducted to demonstrate the potential of Getis–Ord Gi* statistics as a quicker alternative to the clustering techniques that rely on spatially interpolated variables such as the unsupervised ISODATA clustering method. The unsupervised clustering algorithm has been the more familiar delineation method despite the increased availability of EC measurements [57,58].

## 2. Materials and Methods

### 2.1. Study Area

The Fensholt subcatchment (56°00′17″ N 10°06′23″ E) lies in the Norsminde Fjord catchment west of the town of Odder, Denmark (Figure 1a). The subcatchment area is 6.12 km^2^ with 78% currently used for agriculture. This study was carried out in four small artificially drained catchments that were delineated based on the topography (Figure 1b–e), as well as the total areas being drained by a continuous network of drainage pipes that empty into known drain outlets. The drainage catchments are referred to as D1 (Figure 1b), D5 (Figure 1c), D7 (Figure 1d), and D8 (Figure 1e). Soils in the area are primarily loamy tills. A pair of soil profiles from D7 and D8 were classified as a Stagnic Luvic Phaeozem and Gleyic Cambic Phaeozem, respectively. Additional soil auguring in the area supports the generality of this classification for the entire subcatchment. Minor occurrences of peat and gyttja have been recorded in the lowland areas [59,60]. Soil texture is nonetheless largely sandy loam with a few occurrences of clay loam partly due to clay illuviation, which has transported clay vertically through the soil profile or as a product of hydro-topographical gradients in the area [61]. In general, the artificial drainage pipes have been estimated to be located at depths of 1.0–1.5 m with a spacing of approximately 20 m based on participatory mapping with the farmers.

### 2.2. Flow Accumulation

The flow accumulation was calculated through the R package RSAGA in SAGA GIS [62]. A digital elevation model with a spatial resolution of 10 m × 10 m was used and a Triangular Multiple Flow Direction algorithm [63] was employed in the calculation.

### 2.3. Electromagnetic Induction Survey, Data Processing, and Inversion

A DUALEM-21S instrument (Dualem Inc., Milton, ON, Canada), which is a single-transmitter (Tx) multi-receiver (Rx) frequency domain ground conductivity meter operating on a frequency of 9 kHz, was used to carry out the EMI survey. It has a Tx coil located at one end that is shared by two pairs of Rx coils forming horizontal coplanar (HCP) and perpendicular (PRP) configurations. For the HCP configurations, the Tx-Rx separation distances are 1 and 2 m, and for the PRP configurations they are 1.1 and 2.1 m, respectively. The quadrature-phase and in-phase signal responses of the EMI instrument are representative of the EC_a_ and the magnetic susceptibility of the soil [35]. Each of the four coil configurations results in the sensing of different soil volumes with varying depth sensitivities represented in the EC_a_ measurements [64,65,66]. The depth of exploration (DOE) can be defined as the depth at which the signal accumulates 70% of its total sensitivity. When operating at low induction numbers, the depth sensitivity of EC_a_ measurements can be approximated to be a function of coil spacing (S) and array orientation and to be independent of the soil EC for a given frequency [35]. Hence, the DOE for HCP and PRP arrays are 1.6 S and 0.5 S, respectively, when the instrument is placed on the ground [66]. As such, the 1 and 2 m HCP configurations provide measures of EC_a_ for soil volumes reaching depths of 1.6 m and 3.2 m, whereas the 1.1 m, 2.1 m PRP configurations provide EC_a_ to depths of 0.5 m and 1.0 m, respectively.

The instrument was mounted on a sled (0.3 m above the ground) attached to an all-terrain vehicle (ATV) and the survey was performed in August 2014 along parallel transects spaced approximately 8 to 12 m apart. Georeferencing of the data was carried out using a real-time kinematic (RTK) Global Navigation Satellite System (GNSS) with sub-decimeter accuracy. Dedicated data processing and inversion were performed using the Aarhus Workbench software [67] that performs fully nonlinear inversion using the Aarhus Inv code [68]. The data processing involves both automatic and manual steps. Firstly, the automatic data processing was performed to remove negative EC_a_ values, to correct for the offset between the RTK/GNSS setup and the data from the individual channels, and to improve the signal to noise ratio by averaging the data choosing an appropriate sounding distance (3 m) and running mean width (6 m). A careful choice of sounding distance and running mean width are necessary in order not to smear the data generated by the soil variability at hand and to eliminate the redundant information to reduce the computation time for performing inversions. Once the automatic processing was complete, manual inspection of the raw data was undertaken to identify and remove potential noise due to coupling with anthropogenic sources such as buried cables, metal fences, or proximity of the instrument to the ATV when making turns. The changes made in the raw data during the manual processing were integrated back into the averaged data generated through the automatic processing step.

Later, the processed data were inverted using a quasi-three-dimensional spatially constrained inversion algorithm that applies constraints both in-line and cross-line using Delaunay triangulation [68,69]. A homogeneous eight-layer initial model was chosen with depths to the top of each layer being 0, 0.3, 0.6, 1.0, 1.5, 2.0, 2.6, and 3.3 m, respectively, and with an initial EC estimate of 25 mS m^−1^. After the inversion, EC values were extracted for every 0.2 m depth interval to a depth of 1.8 m. Since significant differences in the Eh at the different parts of the subcatchment were observed at depths between 85 to 165 cm from a complementary study [40], the EC values between depths 0.8 to 1.8 m were averaged and subsequently used in our analysis.

### 2.4. Measurement of NO_3_^−^ Concentrations

In the four drainage catchments, 7 to 18 piezometer nest installation points (45 in total) were chosen to represent transects along the expected direction of the groundwater flow and differences in EC. The layout of the piezometers along the topography and average EC per piezometer nest are detailed in [40] and shown in Figure 1. The piezometers were base-sealed PVC pipes with 43 mm internal diameter, 30 cm screens, and a mesh size of 0.5 mm. For each installation point, a piezometer was installed with its screen placed at the midpoint of the most active nitrate reducing zones, which are likely to occur between 85 and 115 cm. The tentative identification of the most active reduction zones was based on the occurrence of redoximorphic features in the form of red-orange mottles distributed over grey sheaths [70] as observed visually in the soil cores from the boreholes. Additional piezometers were installed for each installation point at depths between 125 to 165 cm to represent a possibly different redox environment. Piezometers were completely installed at least four weeks prior to the first collection of the water samples. The specific piezometers and their respective location coordinates and installation depths are listed in Table A1 in Appendix B.

Collection of water samples for NO_3_^−^ analysis was carried out between February and August 2017, typically at intervals of three weeks. Field measurements were planned to succeed a rainfall period to increase the likelihood of collecting a substantial number of water samples. No measurements were possible in July 2017 due to insufficient water levels in the piezometers.

The piezometers were emptied first prior to the collection of the water samples. The time between emptying the piezometer and sample collection varied from six to eight days (February to April 2017) to one to three days (remaining period). Approximately 10 to 300 mL of soil water were present in the piezometers during sample collection. Water samples were filtered on-site using Sefar Nitex 03-31/24 monofilament (Sefar AG, Heiden, Switzerland) with a mesh size of 31 µm. They were kept in 250-mL HDPE bottles at 4 °C until analysis. NO_3_^−^ concentrations were measured within 24 h after sample collection using the colorimetric method through the segmented-flow SEAL Autoanalyzer III (SEAL Analytical, Inc., Mequon, WI, USA).

Despite the possible differences in the nitrogen input between the different drainage catchments, the NO_3_^−^ concentrations presented from each installation point is the average NO_3_^−^ concentrations of the samples within the entire sample collection period (February to August 2017). The average NO_3_^−^ concentrations were used due to the lack of significant differences in the temporal analysis of the values. In addition, only piezometers installed between 85 to 165 cm were considered, since the differences in Eh were most pronounced within this depth range. The NO_3_^−^ concentrations are available as Appendix A [71].

### 2.5. Measurement of Redox Potential Values

Each piezometer had a complementary redox probe installed into the soil, approximately 20 cm from the midpoint of the screen depth. The redox probes are platinum wires with copper tips with the Pt tip in contact with the soil [72]. Prior to installation, the redox probes were checked for accuracy using a hexacyanoferrate II/III redox buffer (pH 7). The redox probes were installed into channels initially made by a stainless steel rod with a 6 mm diameter.

The redox potential was measured by connecting a pH meter to the Cu tip of the probes while a calomel reference electrode was placed approximately 3 cm into the soil. Measured redox values were converted to standard hydrogen electrode potentials (Eh) by adding +245 mV as a correction for the reference calomel electrode. Similar to the average NO_3_^−^ concentrations, the average Eh for the entire water sample period was calculated for each installation point and are available as Appendix A [71].

### 2.6. Spatial Autocorrelation Using Global Moran’s I Statistic

Spatial autocorrelation of the EC values was determined prior to the hotspot analysis using the spatial autocorrelation tool to determine the values’ dependence on its neighbors. The spatial autocorrelation tool utilizes Moran’s I spatial statistics that simultaneously determines the features’ similarity based on their location and values [73]. The Moran’s I statistic is calculated as:(1)$$I=\frac{n}{S_{0}}\frac{\sum_{i=1}^{n}\sum_{j=1}^{n}w_{i,j\;}z_{i}z_{j}}{\sum_{i=1}^{n}z_{i}^{2}}.$$
where $z_{i}$ and $z_{j}$ is the deviation between the features, (i.e., in this case the EC value at positions i and j, respectively, from the mean); $w_{i,j\;}$is the spatial weight between the values at i and j; n is the total number of the features; and $S_{0}$ is the sum of all the spatial weights. The spatial weight, $w_{i,j\;}$, depends on the conceptualization of the spatial relationships between the features. The fixed distance band method was selected for calculating the spatial weights as this was recommended for point data [74]. In the fixed distance band method, features that are within the specified distance band were given an equal spatial weight of 1 and any features outside of it were ignored. A distance band of 14.5 m was used as computed by the spatial statistics tool.

The closer the Moran’s I index value is to 1, the more clustered are the data, while the closer it is to −1, the more dispersed. A value of 0 indicates perfect randomness. The Moran’s I spatial statistics tool also generates the z-score and *p*-value of the Moran’s I index to indicate the statistical significance of the Moran’s I index. A positive z-score indicates that the values are spatially clustered, while a negative z-score indicates that the values are clustered in a competitive way (e.g., high values repelling high values).

### 2.7. Geostastistics of EC

The trends and distribution of the EC values were analyzed first prior to kriging and were normalized through box-cox transformation. Second-order trend removal was applied based on a directional trend analysis of the EC values. To conduct the trend analysis, EC values were plotted on an x,y plane, and its values were projected as stick heights on a z-plane. By fitting polynomial equations through the scatterplots, the directional trend across the x,z and y,z planes may be determined. The trend analysis and the proceeding geostatistical analyses were carried out in ArcGIS 10.0 software (ESRI, Redlands, CA, USA).

To map EC, kriging was used to predict the values for locations without measurements. The prediction map was generated within the geostatistical analyst extension module. Ordinary kriging was used to generate the prediction map of EC values. The equation for ordinary kriging is:(2)$$\hat{z}\left(s_{0}\right)=\sum_{i=1}^{n}\lambda_{i}z\left(s_{i}\right)$$
where $\hat{z}\left(s_{0}\right)$ is the predicted value for the location $s_{0}$; $z\left(s_{i}\right)\;$is the measured value at location $s_{i}$; $\lambda_{i}$ is the interpolation weight for the measured value at the location i; and n is the number of neighboring locations searched for the interpolation. The value of $\lambda_{i}$ will depend on the semivariogram model which is a function for the semivariance against the distance between the observation points. The semivariance was calculated for each predicted value as follows:(3)$$\gamma \left(s_{i}{,s}_{j}\right)=\frac{1}{2n}\sum {[z\left(s_{i}\right)-\;z\left(s_{j}\right)]}^{2}$$
where $\gamma \left(s_{i}{,s}_{j}\right)$ is the semivariance between measurements at $s_{i}$ and $s_{j}$; $z\left(s_{i}\right)-\;z\left(s_{j}\right)$ is the difference between the predicted values $z\left(s_{i}\right)$ and $z\left(s_{j}\right)$; and n is the number of locations considered. The semivariances are plotted against the distance between the measured values. A semivariogram model is fitted into this plot to determine the value of $\lambda_{i}$. The semivariogram model that most closely satisfied the criteria of $\sum_{i=1}^{n}\lambda_{i}=1\;$within the location of the n observations was chosen. The models were evaluated based on the error values between the measured and predicted values through the mean error (ME) and standardized root mean square error (RMSE) of the model. Ideally, ME and RMSE would be close to 0 and 1, respectively.

### 2.8. Clustering of EC Values with Optimized Hot Spot Analysis (Getis–Ord Gi* Statistics)

The Optimized Hot Spot Analysis was carried out by calculating the Getis–Ord Gi* statistics (z-score) for each EC value. The calculation of the Getis–Ord Gi* z-score is:(4)$$G_{i}^{*}=\frac{\sum_{j=1}^{n}w_{i,j}x_{j}\overset{X}{-}\;\sum_{j=1}^{n}w_{i,j}}{S\sqrt{\frac{n\;\left[\sum_{j=1}^{n}w_{i,j}^{2}-{\left(\sum_{j=1}^{n}w_{i,j}\right)}^{2}\right]}{n-1\;}}}$$
where $x_{j}$ is the attribute value for feature j, $w_{i,j}$ is the spatial weight between feature i and j, n is equal to the total number of features and
(5)$$X=\frac{\sum_{j=1}^{n}x_{j}}{n}$$
and
(6)$$S=\sqrt{\frac{\sum_{j=1}^{n}x_{j}^{2}}{n}}-{\left(\;\bar{X}\right)}^{2}$$

The local sum of the z-scores of the EC values is compared with the expected local sum. If the difference between the calculated and expected local sum is too large and less likely to be a result of a random chance, the z-score of the feature is statistically significant. Similar to the Moran’s I statistic, the fixed distance band method was selected for calculating the spatial weights, $w_{i,j}$. A *p*-value is also generated along with the z-scores.

A high EC value with a highly positive z-score and a low *p*-value indicates that the particular value is significantly surrounded by other high EC values. Such EC value is labeled as a ‘hot spot.’ Alternatively, a ‘cold spot’ would have a highly negative z-score and a low *p*-value.

With Optimized Hot Spot analysis, the EC values were automatically assessed whether changes to the data (e.g., aggregation) have to be performed before analysis. Scale distance determination through analysis of incremental spatial autocorrelation was also initially performed. False Discovery Rate (FDR) correction method was also applied to the z-scores to resolve possible issues with multiple testing and spatial dependence. The issue with multiple testing arises when a high number of data would increase the possibility of producing false results (i.e., type 1 error) during comparison of the EC values with each other [75]. A comparison of the significance of a feature based on the context of neighboring features would also exacerbate the influence of spatial patterns that in actuality may only be minimally present or even non-existent. Hence, the issue with spatial dependence may also arise [74].

The FDR correction method ranks the significant *p*-values according to the smallest to largest values, then removes the smallest values according to an estimate of the number of false positives based on the confidence level. The remaining EC values with significant *p*-values are binned together according to specific confidence levels [74]. The specific confidence levels associated with the binned *p*-values are referred to in this paper as ‘Gi bin’. The confidence levels have values ranging from −3 to 3, wherein −3 to -1 is classified as a ‘cold spot’ area; −1 to 1 as ‘not significant’; and 1 to 3 as ‘hot spot’ area [74]. The cold spot and hot spot features are further classified according to confidence intervals of 99%, 95%, and 90%. The higher the absolute value of the confidence level, the higher is its significance level, i.e., a confidence level of -3 indicates that it is a 99% cold spot while a value of 3 indicates it is a 99% hot spot. Similarly, a confidence level with an absolute value of 2 and 1 reflects 95% and 90% confidence intervals, respectively.

### 2.9. Clustering of EC Values Using Unsupervised ISODATA Clustering

To delineate clusters in the EC prediction map, an unsupervised ISODATA clustering algorithm [43] was applied using the Geostatistical Wizard module (ESRI, Redlands, CA, USA). It is a highly heuristic and iterative process which seeks for the minimum Euclidean distance between a candidate cell (pixel) to a cluster [74]. Cluster centers are randomly placed when calculating the Euclidean distance during each iteration. Within each cluster, the standard deviation is calculated from which the merging and splitting of clusters will be based upon. Clusters are merged either if the distance between two clusters is less than the defined threshold or if the number of cells in a cluster is less than the set minimum. A cluster is otherwise split if the standard deviation is higher than the threshold value or if the number of cells is twice the threshold for the minimum number of cells [43]. The process of randomly creating cluster centers and calculating standard deviation is repeated until the maximum number of iterations is reached. The default clustering parameters in the Iso Cluster Unsupervised Classification tool were used wherein the minimum number of cells is 20, sample interval is 10, and number of iterations is 20. Three classes were set for clustering EC values to make it comparable with the clusters from the hotspot analysis described in Section 2.8.

### 2.10. Data Analysis

Statistical analyses were performed using SigmaPlot 11.0 (Systat Software, Inc., San Jose, CA, USA). A two-way ANOVA was conducted to test if the resulting classes generated from each clustering method had an effect on the NO_3_^−^ and Eh values. A Dunn’s post hoc test was conducted to determine significant differences between specific classes following a normality test on the data.

## 3. Results and Discussion

### 3.1. Distribution and Descriptive Statistics and Spatial Distribution of EC Values

The highest EC values were mostly found in the eastern part of the subcatchment, particularly in the D1 drainage catchment and in discrete spots in the D5 drainage catchment (Figure 2). The histogram of the data (Figure 3) showed that values (*n* = 5693) were slightly skewed to the left, which indicates a non-normal distribution of the data due to several relatively high EC values ranging approximately between 60 and 100 mS m^−1^ which are located in D1 drainage catchment (Figure 2). Several portions of the drainage catchment that do not have EC_a_ measurements are either built-up or inaccessible areas where measurements could not be physically conducted (Figure 2).

The spatial variability of the EC values was best described by a tetraspherical variogram model (Figure 4) wherein the predicted EC values have an ME of <0.01, and RMSE of 1.0. Figure 4 lists the fitted parameters for the model. The lag distance of 9 m confirms that the minimal distance of the survey lines is the optimal distance for collecting the EC_a_ measurements. Given that a nugget ratio of 0.25 or less indicates a strong spatial autocorrelation [76,77], a 0.20 ratio from the semivariogram indicates a strong spatial autocorrelation of EC values within the range of 99 m. The small effective range of 99 m indicates a distribution pattern of small patches [78,79]. The spatial autocorrelation of the EC values was also confirmed with the results of Moran’s I statistics. A Moran’s I index value of 0.8 with a significant z-score of 105 (*p* < 0.01) shows that there is significant autocorrelation at 99% confidence within a distance of 14.5 m, which is slightly higher than the maximum survey transect line distance (i.e., 12 m).

The significant spatial autocorrelation of the EC values confirms that the clusters of EC values are determined by consistent, systematic processes that occur on a small scale [80]. Given that the EC is significantly correlated with clay content in non-saline agricultural soils [81], it is expected that the variations in the EC values are mainly due to the heterogeneity in the clay contents. Indeed, clay illuviation zones were observed at depths approximately 50 to 100 cm above a relatively impermeable layer under a calcareous boundary [61]. Zones of clay accumulation in the area most likely consist of inherent zones of clay layers of lenses due to the natural geological variations in the soil and/or mobilized clay particles transported through hydro-topographical gradients in the area [61], which explains the high clay content even at depths between 25 to 75 cm in low elevated areas [82]. Indeed, EC had been demonstrated to be significantly correlated with the clay content measured in soil samples collected at varying locations and depths [40], which suggests that EC was largely due to surface conduction [83] in the subcatchment. The association of high clay percentage with high EC was considered to enable the presence of anoxic microsites that led to the more reducing conditions in these areas, thereby rendering them as potential NRH areas [40].

Based on the parameterization of three different modeling approaches by [61], the horizons with clay illuviation (90 to 130 cm) were estimated to have a range of saturated hydraulic conductivities between 0.04–0.36 m h^−1^. In contrast, saturated hydraulic conductivity at soil depths of 0 to 25 cm had a range of 0.12–0.62 m h^−1^ [61]. The low saturated hydraulic conductivity slightly above the drainage pipes seems to promote a perched water table [40,84].

In addition to the soil geochemical heterogeneity, topography also influences the distribution of water in the soil, as demonstrated by the high flow accumulation values, wherein the shallow groundwater tends to flow towards the least elevated portions (Figure 1b,e). Topography and clay content were, therefore, two prominent factors that influence the variability of the water content in the respective catchments. The topographical gradients in combination with low hydraulic conductivity of areas with high clay contents, regardless of whether the clay is inherent or due to accumulation over time, controls the overall soil water distribution, subsequently resulting in the high EC values in certain distinct areas in D1 and D5 [17,27].

### 3.2. Clustering of EC Values

#### 3.2.1. Optimized Hot Spot Analysis of EC Values (Getis–Ord Gi* Statistics)

Figure 5 summarizes the statistics of the EC values from each class generated through Optimized Hot Spot Analysis. Though the mean EC values progressively increase from the cold spot to hot spot classes, there is an overlap in the range of the EC values between classes. The mean EC values for cold spots ranged from 27 to 33 mS m^−1^, while the range is 39 to 49 mS m-1 for the hot spot areas. Areas classified as ‘not significant’ have a mean EC value of 36 mS m^−1^ (Figure 5).

A total of 30% of the points were classified as hot spots; 33% as not significant; and 37% are cold spots. A majority of the points classified as either a hot spot or cold spot are in the 99% confidence level (Figure 5), which indicates sharp boundaries between the clusters of EC values. The hot spots were mainly found in the D1 drainage catchment and in discrete portions in D5 where groups of high EC values were found (compare Figure 2 and Figure 5). Despite several high EC values in D7 and D8 (Figure 2), these were only areas registered as ‘not significant’. A majority of the areas in D7 and D8 were ‘cold spots’ (Figure 5).

#### 3.2.2. Unsupervised ISODATA Clustering Results

Figure 6 shows the results of ISODATA clustering on the EC values. Class 1 had the lowest mean EC estimate (27 mS m^−1^), while Class 3 had the highest mean estimate (49 mS m^−1^), and Class 2 had an intermediate value (36 mS m^−1^). Most of the areas were classified as Class 2, which occupied 43% of the total drainage subcatchment area in the study.

Based on their location and the corresponding statistics of the EC values, the generated classes closely corresponded with the classes generated through Optimized Hot Spot Analysis. Class 3 from the unsupervised ISODATA clustering corresponded to the ‘hot spot’ areas according to the Optimized Hot Spot Analysis of EC values (compare Figure 5 and Figure 6). Similarly, Class 2 corresponded to ‘not significant’ areas, and Class 1 corresponded to ‘cold spot’ areas (compare Figure 5 and Figure 6). Nonetheless, there were still discrepancies in the area classification at a finer scale. The high EC values in the D8 drainage catchment were classified as Class 3 (Figure 6) though these were classified as ‘not significant’ through Optimized Hot Spot Analysis (Figure 5). Similarly, Class 2 areas surrounded with Class 3 areas in D1 (Figure 6) were altogether classified as part of the ‘hot spot’ area through Optimized Hot Spot Analysis (Figure 5). Subzones generated through Optimized Hot Spot Analysis were generally more homogenous than subzones generated through unsupervised ISODATA clustering. The more fragmented clusters generated through the unsupervised ISODATA clustering method were likely due to the low minimum number of cells that was set per cluster. The standard deviation may also have been easily achieved even with a small number of cells in a cluster. The unsupervised ISODATA clustering had been observed to be quite sensitive to clustering parameters [85,86]. Nonetheless, a t-test analysis comparing the percentages of the total area between corresponding classes show that they are not statistically significant (*p* = 1.0).

### 3.3. Comparison of Redox Potential Values and NO_3_^−^ Concentrations from Classified Piezometers

Figure 7 shows the Eh and the NO_3_^−^ concentrations from the piezometers that were classified based on the clusters generated through Optimized Hot Spot Analysis (Figure 5) and the unsupervised ISODATA clustering (Figure 6). A separate ANOVA analysis of the Eh and NO_3_^−^ concentrations between the clusters generated through both clustering methods shows that the values are statistically significantly different from each other (*p* < 0.001). For the clusters from Optimized Hot Spot Analysis (Figure 5), a post-hoc Dunn’s test analysis shows that the Eh and NO_3_^−^ concentrations from the ‘hot spot’ areas (Figure 7a,c) were significantly lower than in areas classified as ‘not significant’ and ‘cold spot’ (*p* < 0.05). No significant differences were found between the values of ‘cold spot’ and ‘not significant’ areas. Similar results were also found in the statistical analyses of the Eh and NO_3_^−^ concentrations from the classes generated through unsupervised ISODATA clustering (Figure 7b,d). Considering their significantly lower Eh and NO_3_^−^ concentrations, areas classified as either Class 3 or ‘hot spot’ using the aforementioned clustering methods could be considered as NRH. As the range Eh for the Class 3 and hot spot areas was −129 to 321 mV and −129 to 282 mV, respectively, it is possible that other reduced cations, such as Fe^2+^, are present, given that nitrate is fully reduced below 100 mV [85].

The lack of significant differences between the Eh and NO_3_^−^ concentrations between classes other than ‘hot spot’ and ‘class 3′ (Figure 7) indicate that significant differences in the EC values may not necessarily demonstrate significant differences in the nitrate reduction processes. It is however likely that the projected small-scale heterogeneity in clay content and water saturation, as indicated by EC values, essentially influences the redox conditions and NO_3_^−^ concentrations in the NRH [80,87]. Therefore, the NRH may still be effectively delineated from the drainage subcatchments. Other than clay content, NRH areas were also associated with high flow accumulation values which indicate the likelihood that they have the potential to also reduce NO_3_^−^ from the incoming shallow groundwater from the more elevated portions.

Appendix B lists the classification of the piezometer locations according to the unsupervised ISODATA clustering and Optimized Hot Spot Analysis. A comparison of the classifications shows that there was a 16% inconsistency between the two clustering methods (Figure 8c). Most of the inconsistencies were due to several piezometers classified as ‘Class 2′ being considered as ‘cold spots’ instead of ‘not significant’ areas through the Optimized Hot Spot Analysis. Piezometers with inconsistent classifications were only observed in D1 (Figure 8a) and D7 (Figure 8b) and were located close to the borders of the classes, which likely indicates transition areas from one class to another. Nonetheless, the lack of significant differences in the Eh and NO_3_^−^ concentrations between Class 1 and Class 2, and between ‘cold spot’ and ‘not significant’ areas means that the discrepancy between these two classifications is of minor importance in terms of delineating NRH, which is the main focus of our work. In spite of the lack of significant differences in the Eh and NO_3_^−^ concentrations between the non-NRH classes (Figure 7), the spatial heterogeneity in clay and water content, as observed through significant differences in the EC values, may still have an influence on the nitrate reduction mechanism, and should still be considered for future studies in the area.

Despite the discrepancy in the classification of the piezometers, both clustering methods have comparable mean EC values (compare Figure 5 and Figure 6), range of Eh and NO_3_^−^ concentrations (Figure 7), and the total percentage of points/area per class (compare Figure 5 and Figure 6). The classification according to the Optimized Hot Spot Analysis complements well the results of the unsupervised ISODATA clustering but has the additional advantage of presenting the transitions between the borders (Figure 5), which tends to be an issue with unsupervised ISODATA clustering [45]. Identification of these transition areas around the boundaries, coupled with the flow accumulation data (Figure 1b–e), may be considered when planning further sample collection and nitrogen reduction studies, particularly in the context of estimating the location and extent of nitrate reduction processes [88]. The layout of the subzones generated through Optimized Hot Spot analysis was also less fragmented and more homogeneous (Figure 5) [89]. It must be stressed, however, that the feasibility of generating comparable results between unsupervised ISODATA clustering and Optimized Hot Spot Analysis could be due to the small scale and high spatial autocorrelation and density of the EC values. Considering these additional advantages, the use of Optimized Hot Spot Analysis is more advantageous in delineating possible NRH areas based on the EC values.

Optimized Hot Spot Analysis, which has been gaining interest and use during the last 10 years, could be a straightforward technique to utilize available EC values derived from EMI sensors. It is a promising clustering technique that can be used for an exploratory delineation of NRH in other artificially drained agricultural areas.

## 4. Conclusions

Soil electrical conductivity (EC) values derived from the inversion of apparent electrical conductivity data measured with an electromagnetic induction instrument proved to be an effective means for identifying nitrate reduction hotspots (NRH) in artificially drained catchments. The NRH areas may be delineated either through the use of Getis–Ord Gi* statistics (utilized through Optimized Hot Spot Analysis) or unsupervised ISODATA clustering of the average EC values at depths between 80 to 180 cm. The significant clustering of EC values at a small scale indicates the presence of underlying spatial heterogeneity that promotes differences in their average redox potential (Eh) values and NO_3_^−^ concentrations. Areas with significantly clustered high EC values, consequently classified as NRH areas, were associated with high flow accumulation controlled by the topography as well as impeding clayey soil layers that likely increases the water saturation values to induce intense nitrate reduction. Though the generated clusters align from both clustering techniques in terms of their average EC values, Eh values, and NO_3_^−^ concentrations, Optimized Hot Spot Analysis has the additional advantage of presenting the transition between subzone borders and has less fragmented clusters. The average NO_3_^−^ concentration from NRH areas is 2.0 ± 2.2 mg/L, which is significantly lower than the concentrations from non-NRH areas. Approximately 26% of the drainage catchments in the study may be considered as NRH areas, suggesting that the estimates of the nitrate removal in the vadose zone may be underestimated at the subcatchment level. The determination of the location and extent of the NRH would be extremely beneficial in supporting site-specific strategies for nitrogen management across different scales.

## Figures and Tables

**Figure 1 sensors-22-01508-f001:**
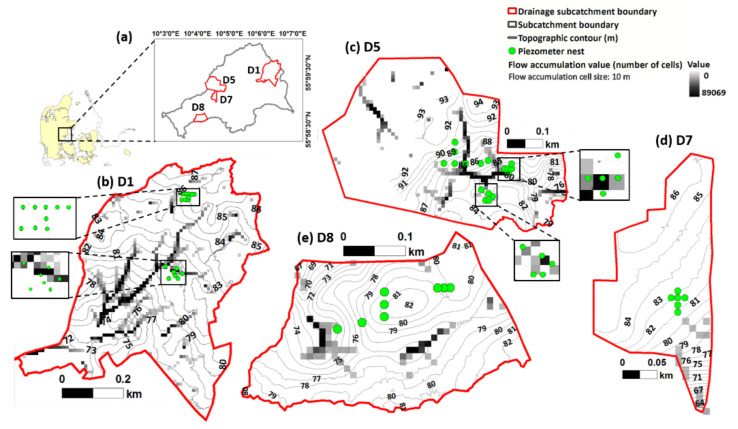
(**a**) Fensholt subcatchment and the location of the drainage catchments, (**b**) D1, (**c**) D5, (**d**) D7, and (**e**) D8, and their corresponding topographic elevation contours, flow accumulation values, and piezometer nests within the boundaries. Each piezometer nest consists of 1 to 3 piezometers.

**Figure 2 sensors-22-01508-f002:**
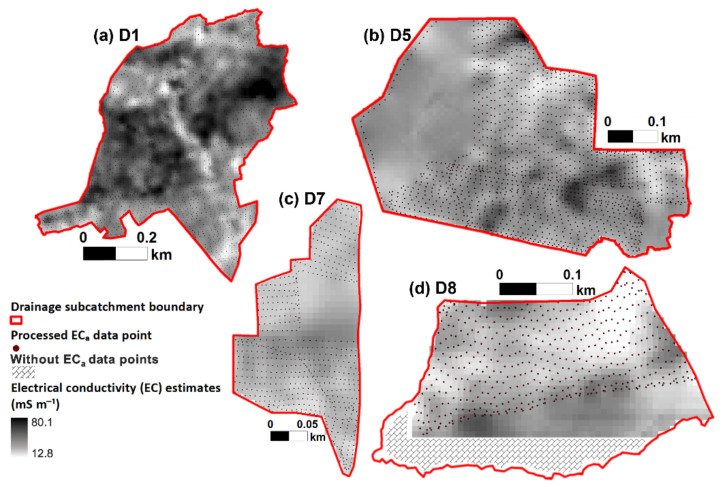
Spatial distribution of the electrical conductivity (EC) values in drainage catchments (**a**) D1, (**b**) D5, (**c**) D7, and (**d**) D8 using the ordinary kriging method. The average of the EC values between the depths of 80 to 180 cm was used for the interpolation. Portions without EC are either built-up or inaccessible areas where physical measurements cannot be conducted.

**Figure 3 sensors-22-01508-f003:**
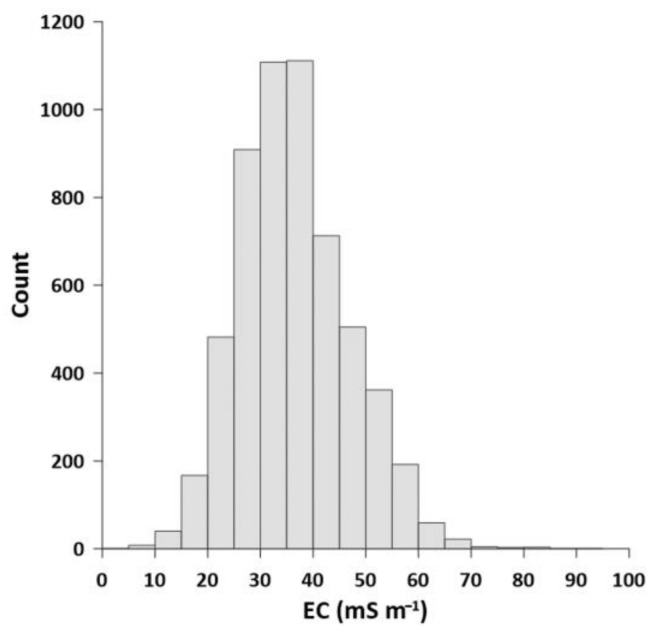
Histogram of the average electrical conductivity (EC) values at 80 to 180 cm depth from the four drainage catchments.

**Figure 4 sensors-22-01508-f004:**
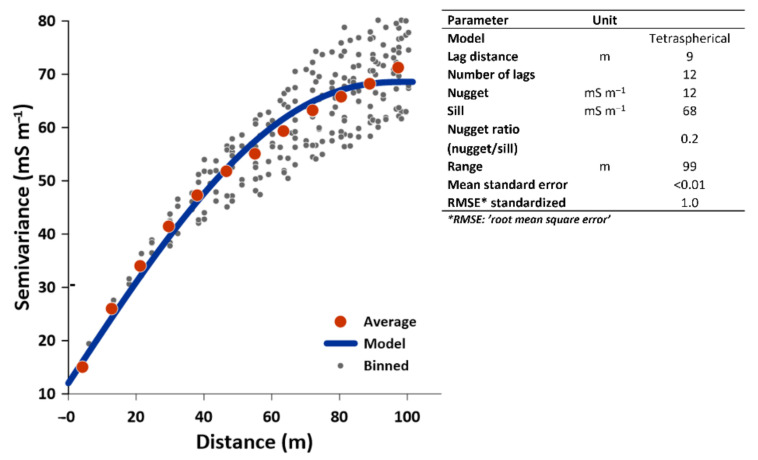
Semivariogram and the values of the fitted parameters for the prediction model of the electrical conductivity (EC) values. A tetraspherical model was used for its generation.

**Figure 5 sensors-22-01508-f005:**
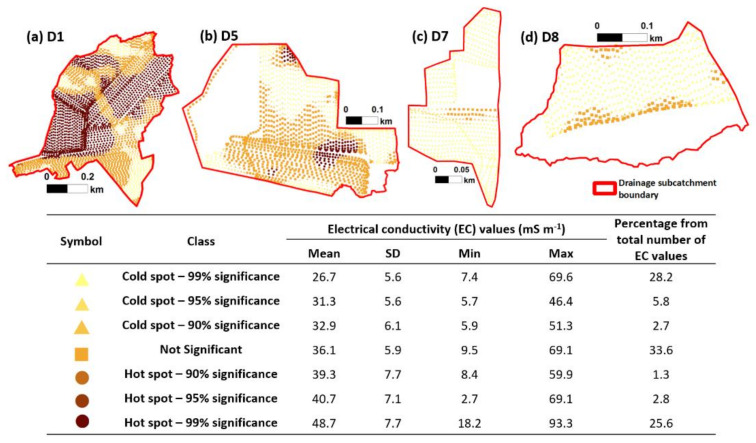
The layout of the clusters generated from the Optimized Hot Spot Analysis of average electrical conductivity (EC) values across the drainage catchments (**a**) D1, (**b**) D5, (**c**) D7, and (**d**) D8. The accompanying table lists the mean, standard deviation (SD), minimum EC value (min), maximum EC value (max), and the percentage of the number of EC values per each generated class.

**Figure 6 sensors-22-01508-f006:**
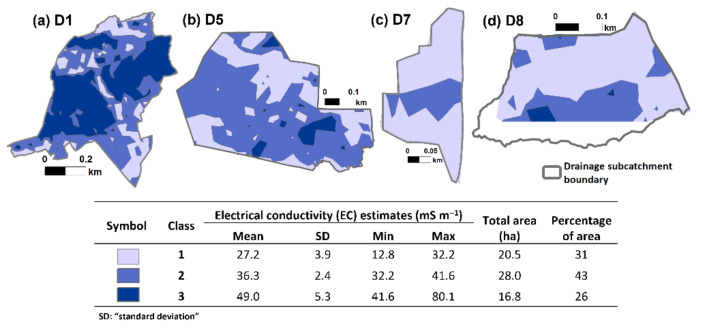
The layout of the classes generated from the unsupervised ISODATA clustering of average electrical conductivity (EC) values across the drainage catchments (**a**) D1, (**b**) D5, (**c**) D7, and (**d**) D8. The accompanying table lists the mean, standard deviation (SD), minimum EC value (min), maximum EC value (max), and the percentage of the area of each generated class. Class 3 has the highest average and range of EC values.

**Figure 7 sensors-22-01508-f007:**
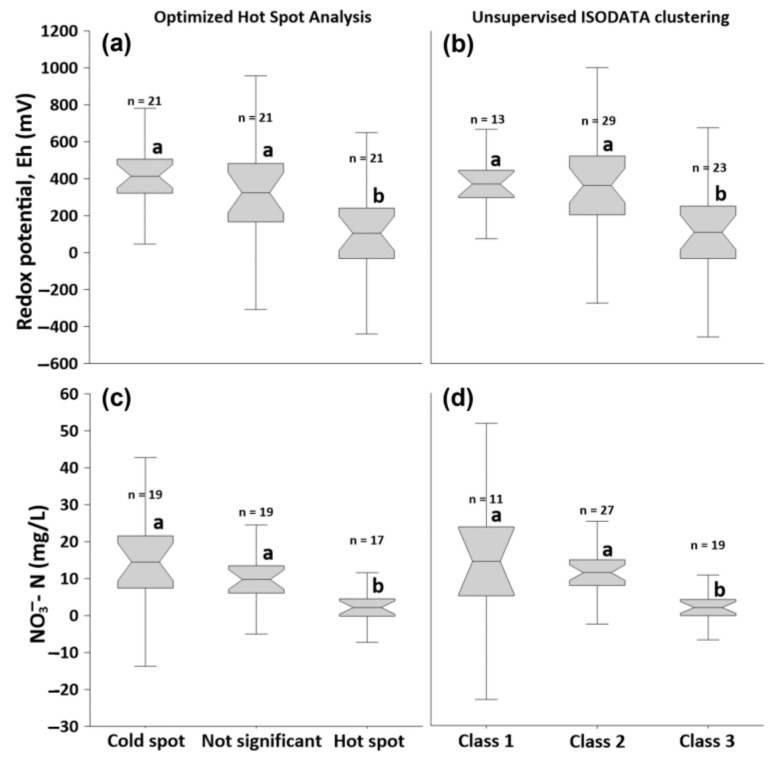
Boxplots of (**a**,**b**) average redox potential values (Eh) and (**c**,**d**) NO_3_^¬^- concentrations and from each of the classes generated through (**a**,**c**) Optimized Hot Spot Analysis and (**b**,**d**) unsupervised ISODATA clustering of EC estimates. The midline for each of the boxplots is the mean, the box edges are the standard deviation, and the whiskers are the interquartile range (IQR) with a factor of 1.5. The notch (i.e., narrow interval of the box) factor is 1.7, which is approximately 95% confidence in the median. Boxplots sharing the same letter are not significantly different (*p* < 0.05) according to post-hoc Dunn’s test analysis.

**Figure 8 sensors-22-01508-f008:**
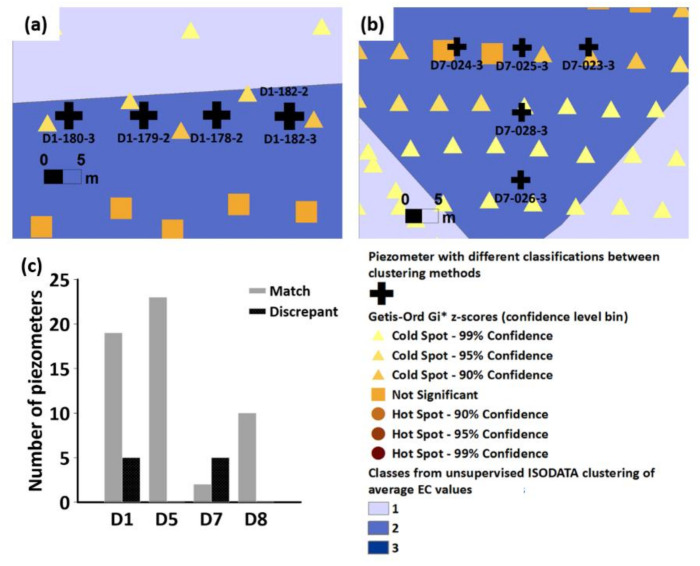
Location of piezometers (✚) in drainage catchments (**a**) D1 and (**b**) D7 with discrepant classifications based on the clusters generated through Optimized Hot Spot Analysis and unsupervised ISODATA clustering method. (**c**) Comparison of the piezometer classifications under both clustering methods shows that a total of 10 piezometers had discrepant classifications. No discrepant classification was found in drainage catchment D5 and D8. For the detailed list and locations of the piezometers and their corresponding classifications, please see Appendix B.

## Data Availability

Data is contained within the article. Appendix A is published as Mendeley Data available at http://dx.doi.org/10.17632/235yhyjjbt.1 (accessed on 1 September 2020).

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
