# Peer review of "Delineation of Nitrate Reduction Hotspots in Artificially Drained Areas through Assessment of Small-Scale Spatial Variability of Electrical Conductivity Data"

_sensors, 2022, doi:10.3390/s22041508_

Round 1

Reviewer 1 Report

The paper is interesting piece of work and the author tries to delineate nitrate reduction hotspots in artificially drained areas through assessment of small-scale spatial variability of electrical conductivity data. But the methodologies described and results obtained lack clarity and more work needs to be done before publishing the scientific findings in this journal.

Few of my observations are given below:

  1. Experimental procedure should be clearly described.
  2. Number of data and data structure is not clearly stated. To establish any scientific result, number of data sets should be statistically significant.    
  3. Optimization of hotspot is not described properly.
  4. How has Global Moran’s I statistic been calculated based on EC values?
  5. Basis of identification of hotspot is not clear. Though it is calculated based on Getis-Ord CI statistics but needs more elucidation.
  6. What is the Type 1 error during comparison of the EC values?
  7. Post-hoc Dunn’s test analysis needs clarifications with this data sets.

So, based on the above mentioned points, the manuscript may be revised.

Reviewer 2 Report

The authors have tried to study an interesting and important topic on delineating nitrate reduction hotspots. The authors are required to start the "Introduction" section by giving a brief introduction on 'nitrate reduction hotspots' and how/why studying them is important. The next paragraph can suggest the importance of this kind of study and where the present study is contributing to the scientific knowledge.    The attempt for deriving 'nitrate reduction hotspots' from the apparent electrical conductivity of the soil is an interesting attempt by the authors here.    Is ISODATA an abbreviated term? Please mention its expanded form in the very first mention of ISODATA.   Kindly mask the inaccessible/unmeasured area in the D8 drainage catchment with some other color.   The methods are explained well, however, a lack of cohesiveness in the writing was found in the manuscript.   The methodology was overall simple and the availability of such closely located data points was an advantage. Is there any chance, the authors can suggest in the Conclusions section, what other methods can be used when data scarcity is there for a similar study? Please mention the future scope of such research studies too.   Overall, the work is interesting and of real importance. Minor changes in sentence structures are required for cohesiveness in the writing. 

Round 2

Reviewer 1 Report

The authors have addressed the review comments adequately in the revised manuscript. Hence, recommend acceptance of the paper.